# Fast Aero-Structural Model of a Leading-Edge Inflatable Kite



**Oriol Cayon** [1,2] **, Mac Gaunaa** [1] **and Roland Schmehl** [2,*]

1   Department of Wind Energy, Technical University of Denmark, 4000 Roskilde, Denmark;
    o.cayon@tudelft.nl (O.C.); macg@dtu.dk (M.G.)
2   Faculty of Aerospace Engineering, Delft University of Technology, 2629 HS Delft, The Netherlands
*   Correspondence: r.schmehl@tudelft.nl

**Abstract:** Soft-wing kites for airborne wind-energy harvesting function as flying tensile membrane structures, each of whose shape depends on the aerodynamic load distribution and vice versa. The strong two-way coupling between shape and loading poses a complex fluid–structure interaction problem. Since computational models for such problems do not yet meet the requirements of being accurate and at the same time fast, kite designers usually work on the basis of intuition and experience, combined with extensive iterative flight testing. This paper presents a fast aero-structural model of leading-edge inflatable kites for the design phase of airborne wind-energy systems. The fluid–structure interaction solver couples two fast and modular models: a particle system model to capture the deformation of the wing and bridle-line system and a 3D nonlinear vortex step method coupled with viscous 2D airfoil polars to describe the aerodynamics. The flow solver was validated with several wing geometries and proved to be accurate and computationally inexpensive for pre-stall angles of attack. The coupled aero-structural model was validated using experimental data, showing good agreement in the deformations and aerodynamic forces. Therefore, the speed and accuracy of this model make it an excellent foundation for a kite design tool.

**Keywords:** airborne wind energy; fluid–structure interaction; vortex step method; lifting line method; particle system model; membrane structures

## 1. Introduction

Wind is one of the few renewable resources that will be able to meet the future global energy demand [1]. Most of this resource is found at higher altitudes, where the wind is stronger and more constant. However, conventional wind turbines are limited to relatively low altitudes, hence the great appeal of airborne wind energy (AWE) systems, which use tethered flying devices to access higher altitudes. In this way, between 70 and 90% of the materials required for conventional wind turbines with the same energy output can be saved in large-scale deployments [2–4]. To this end, several approaches have been explored, including the concept of Kitepower B.V. using a soft-wing kite to pull a tether from a ground-based drum driving a generator. The flexible membrane wing is used as a morphing control surface, actuated by modifying the geometry of the bridle-line system via adjustments in the line lengths [5]. Due to the high degree of flexibility of such kites, its shape strongly depends on the aerodynamic force distribution, and in turn, the flow around the wing is affected by the changing shape. Consequently, it is necessary to consider both the actuation system and the fluid–structure interaction (FSI) problem to study the aero-structural behavior of kites.

The typical approach for simulating the FSI problem is to couple a finite element method (FEM) with a computational fluid dynamics (CFD) solver and employ a robust mesh deformation method [6]. However, the computational effort of such a high-fidelity model makes it unfeasible for design optimization, where many configurations have to be evaluated. Simpler and faster aero-structural models for soft kites use lower-fidelity structural and aerodynamic methods. An example is the kite design toolbox presented in [7]

that approximates the tubular frame of the leading-edge inflatable (LEI) kite by a multi-body model and the canopy by a matrix of spring-damper elements, and derives the aerodynamic load distribution from a look-up table of discretized airfoil polars precomputed with CFD. The tool was developed with the commercial, closed-source multi-body solver, MD Adams, using a scripting layer to implement the deformation-dependent load correlation. In [8–11], this correlation was combined with low-resolution FE models of the LEI kite, using beam and shell elements. The lower fidelity aerodynamic models in particular have only limited validity. Due to the low aspect ratio and large curvature, the flow is often governed by strong 3D nonlinear effects that these models cannot capture. Lifting line methods (LLM) using viscous airfoil polars were used in [12–14] to study these effects on the aerodynamics of the static wing shape. In [15], the AeroDyn module of OpenFAST [16] was extended by an LLM for arbitrary wind-energy concepts, including soft kites. In [17], a vortex panel method was coupled to an FE model of a ram-air wing to compute the deformed shape of the wing during flight. The use of the inviscid flow solver resulted in a substantial speed-up compared to the use of a CFD solver in the reference study [6]. Similarly to the LLM, the panel method is also limited to angles of attack well below stall. In [18,19], a beam element model of an LEI kite was proposed. It uses FE analysis to precompute the mechanical properties of the canopy spanning the wing segments. On the other hand, a too-aggressive simplification of the FSI problem to achieve close to real-time simulations, such as in [20], does not allow reproduction of the governing aerodynamic and structural dynamic phenomena.

To our knowledge, there is no aero-structural model available that is suitable for early design stages and optimization processes, due to long computation times, complex simulation setups, limited accuracy, or poor numerical robustness. As a solution to this problem, the present paper proposes an aero-structural model that can be used in the design stage to optimize the geometry of an LEI kite and its bridle-line system. The paper is structured as follows. Section 2 describes the computational approach to simulating the FSI problem. In Section 3, the computed results are presented and compared to reference studies and experimental data. In Section 4, the results are discussed and conclusions are drawn.

## 2. Computational Approach

The LEI V3 kite—originally developed by TU Delft and kite designer Martial Camblong of Genetrix in 2012, and later improved by Kitepower B.V., Delft, The Netherlands—was selected for this study. With the availability of measurement data and video footage from many flight test campaigns, the V3 kite has become a reference design for many computational studies [5,11,21–24]. This kite is depicted in Figure 1, including a front view that assumes the kite is moving towards the reader and a side view that assumes the kite is moving to the right. The kite is composed of an inflatable leading-edge tube in the spanwise direction and inflatable strut tubes in the chordwise direction. The tubular frame forms the structural skeleton of the kite. It is used to support the canopy, a thin fabric membrane, transmitting the distributed aerodynamic load to the bridle lines that connect to the leading edge (LE) and trailing edge (TE) of each strut. The front bridle lines transmit most of the force, and the rear bridle lines are used to actuate the wing. For better force distribution, the front bridle lines split into two short lines just before attaching to the strut tubes, as depicted in Figure 2. The flight path of the kite and the generated force are controlled by the kite control unit (KCU), connected to the steering lines by two tapes. The depower tape is used to change the pitch of the kite, whereas the steering tape actuates the kite asymmetrically, allowing it to turn. The actuation setup of the kite is defined by the power ($u_p$) and steering ($u_s$) settings, which go from 0 to 1, i.e., depowered to powered and no steering to maximum steering, although the kite is never steered to its maximum extent during normal operation.

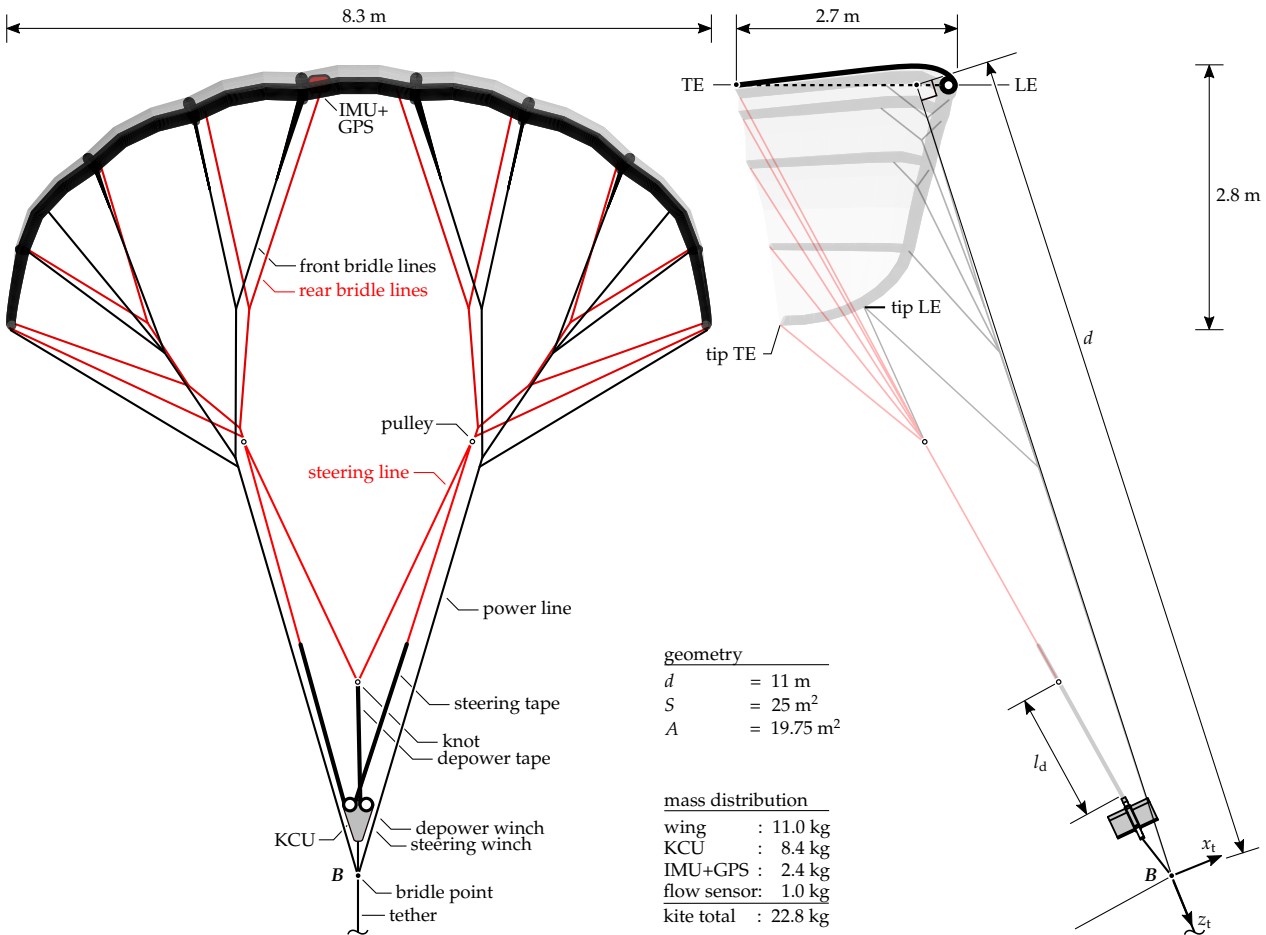

**Figure 1.** Front view (**left**) and side view (**right**) of the LEI V3 kite with geometric parameters, mass distribution, and definition of the reference chord $c_{\text{ref}}$. The total wing surface area is denoted as $S$, and the projected value is denoted as $A$. The mass of the bridle lines is part of the wing mass. The red bridle lines denote the steering lines and the black ones the power lines. The explicit dimensions describe the unloaded design shape of the wing. Adapted from [5].

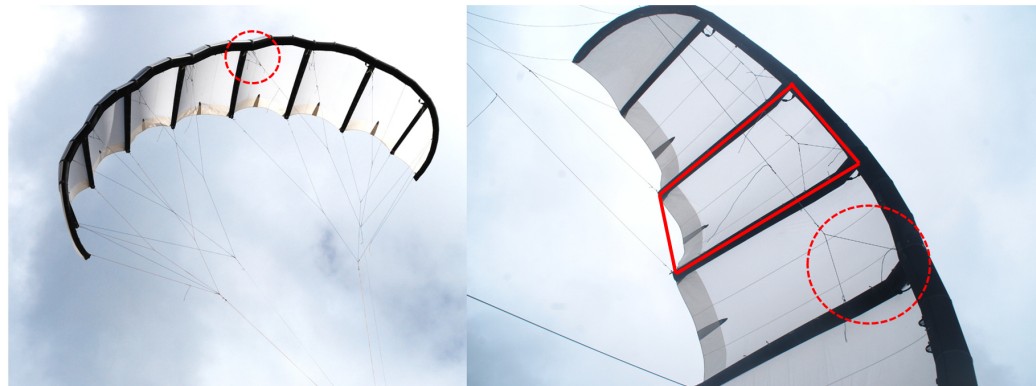

**Figure 2.** Two photos of the LEI V3 kite in flight: the red quadrilateral indicates the structural model of one of the wing segments, and the red circles indicate Y-splits of the bridle close to the tubes to distribute the transmitted load [25].

The aero-structural model is built from independently developed structural and aerodynamic models, intended to be fast and modular, coupled by an algorithm specifically designed for this problem.

### 2.1. Structural Model

To model the deformation of the kite, the assembly of the wing, bridle-line system, and KCU is represented as a set of distributed point masses [25]. This particle system model (PSM) is based on the hypothesis that the shape of the kite is mainly determined by the geometry of the bridle-line system, whereas the aerodynamic forces introduced by the flying membrane structure are mainly responsible for the tensioning of this line system. Sagging of bridle line segments, billowing of canopy segments, and bending of tube segments are considered local FSI effects that can be either neglected or accounted for with specific component-level models.

The wing is discretized into nine segments delimited by the struts. The short Y-splits of the bridle lines close to the tubes are neglected such that each wing segment is supported at four line-attachment points. The approach is illustrated in Figure 2. The particles, which essentially are point masses, are placed at these attachment points and all the knots and pulleys of the bridle-line system, as shown in Figure 3. The connections between the particles are simulated as massless spring-damper elements. The internal forces ($\mathbf{F}_i^s$) acting on each particle *i* from the connected elements *j* are expressed as

$$\mathbf{F}_i^s = -C\dot{\mathbf{x}}_i + \sum_{j=0}^{N_j} \frac{K}{L_j}\epsilon_j\hat{\mathbf{u}}_j, \tag{1}$$

where *C* is the damping coefficient, $\mathbf{x}_i$ is the position vector of each particle, $N_j$ is the number of connected elements, *K* is the element stiffness representing the ratio between the Young's modulus and the cross-sectional area; and $L_j$, $\epsilon_j$, and $\hat{\mathbf{u}}_j$ are the length, elongation, and direction of each connected element.

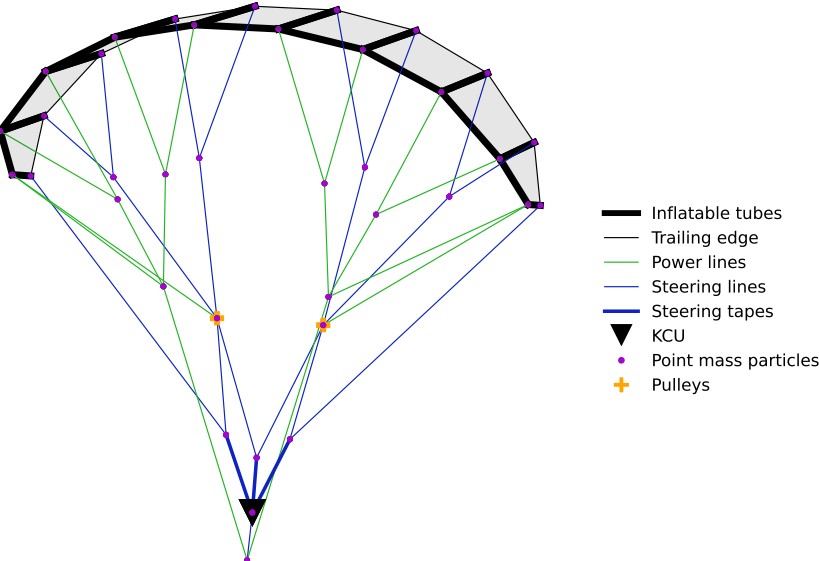

**Figure 3.** Particle-system representation of the LEI V3 kite [26].

Then, the deformation of the kite, i.e., the displacement of the particles, is found by solving Newton's second law for each particle *i*:

$$m_i\ddot{\mathbf{x}}_i = \sum \mathbf{F}_i = \mathbf{F}_i^a + \mathbf{F}_i^s + \mathbf{F}_i^g, \tag{2}$$

where $m_i$ represents the mass of each particle, $\mathbf{F}_i^a$ denotes the aerodynamic force acting at each of the nodes of the nine segments representing the kite, and $\mathbf{F}_i^g$ is the gravitational force. This system of equations is solved using a Runge–Kutta method of the fifth order [25].

The spring properties are chosen to ensure that no significant stretching of any elastic element occurs. Therefore, the shape of the kite is constrained by the geometric properties

of the bridle-line system, which makes this model pseudo-physical. For the specific case study of this kite, the stiffness and damping coefficients are approximately $10^5$ N and $10^1$ kg s$^{-1}$, respectively.

The spring-damper elements used for the structural modeling of wing and bridle lines are assumed to not vary in length except for the elements used for the TE of the wing, which shorten under load due to the billowing of the canopy. This TE contraction effect is taken into account by photogrammetric analysis, relating the power state of the kite to the geometric distances between particles located on the TE. Inflatable tube segments between bridle-line attachment points are assumed to be stiff because the high internal pressure prevents any local bending. Connected tube segments can, however, rotate freely with respect to each other. This assumption is supported by the experimental observation that a pressure drop in the tubes hardly affects the shape of the kite. Consequently, the shape of the kite is mainly dominated by the bridle-line system's geometry, rather than by the structural properties of the kite material and the construction details of the wing [25].

## 2.2. Aerodynamic Model

Despite the dominant influence of the bridle-line system geometry on the wing shape, the PSM requires the distributed aerodynamic loading on the canopy to keep the tensile membrane structure tensioned. For this purpose, a vortex step method (VSM) was developed based on the method described in [27,28], which in turn builds on [29]. The VSM has proven to be robust and computationally inexpensive while still being sufficiently accurate, even for unconventional geometries with low aspect ratios and high anhedral angles.

As in the classic lifting line method (LLM), the chordwise circulation is replaced by a concentrated vortex located at the aerodynamic center of the airfoil, usually at about a quarter of the chord, creating a single filament or lifting line responsible for the generation of lift. According to Kelvin's theorem, the circulation must form a closed loop, and the circulation system is consequently formed by a set of horseshoe vortices that close infinitely downstream.

What differentiates the VSM from the classic LLM is that the control point where the circulation system is solved, i.e., where the magnitude of the aerodynamic forces is calculated, is located at the three-quarter chord position instead of one-quarter of the chord. This is based on Pistolesi's theorem, stating that the angle of attack at the three-quarter chord gives the correct lift associated with vorticity [30]. That being said, there is a subtle but essential difference in the current implementation with respect to [27–29]. Following the conclusions of [31], using thin airfoil theory, it is possible to derive that the three-quarter chord results in the correct position to calculate the magnitudes of the forces and the one-quarter point results in the correct position to calculate the directions of the forces. Therefore, once the magnitude of the circulation is determined using the three-quarter chord angle of attack, the direction in which the local lift and drag forces act can be calculated by analyzing the conditions at the one-quarter chord position—i.e., the local angle of attack can be determined by accounting for the influence of the velocities induced by the vorticity system on the freestream velocity. Once the local lift and drag forces are calculated at the one-quarter chord position, they are transformed into the global lift and drag forces, which are oriented perpendicularly and in parallel to the freestream velocity, respectively. By doing that, a significant improvement is seen in the accuracy of the aerodynamic forces, especially for the drag coefficient, where these methods usually fail and revert to determining the induced drag using the Trefftz-plane approach, which can only be used for steady, straight-flying conditions.

The VSM is coupled to 2D viscous airfoil polars to account for the nonlinear behavior of the lift coefficient, obtained using a correlation model derived from Reynolds-averaged Navier–Stokes (RANS) analysis, detailed in [7]. The analysis was conducted with the Fluent solver, configured as pressure-based, steady, and two-dimensional, using the $k$-$\omega$ SST model for viscous effects. Then, the case was iterated for a range of angles of attack from 0 degrees to 25 degrees, using airfoils with varying thickness ratios and camber,

and the results were used to create a polynomial regression model of the aerodynamic properties of an LEI airfoil.

As the structural model does not consider deformations in the chordwise direction, the camber and thickness of the profiles are kept constant and extracted from the CAD model of the kite. In the same way, the canopy billowing of each panel is also extracted from the CAD model, approximated as a semi-circular arc whose curvature depends on the length of each TE segment. Deriving the semi-circular shape of the canopy between the chordwise struts, detailed in [26], is possible assuming that the fabric is inelastic and that the pressure jump is constant for each panel.

The VSM model is further detailed by the flowchart presented in Figure 4, describing the model's internal logic, structured as follows:

1. Generate the wing geometry, along with the definition of the vortex filaments, control points, and the relevant vectors for each section.
2. Start with an initial guess for the bound circulation ($\Gamma$) to initiate the iterative process to find a solution.
3. Calculate the relative velocity ($\mathbf{U}_{rel}$) at each control point ($P_j$), i.e., the velocity seen by the airfoil, which is found by relating the inner 2D region (airfoil) to the outer 3D region (vorticity system) as follows:

$$\left. \begin{array}{l} \text{2D}: \mathbf{U}_{rel} + \mathbf{U}_{ind,2D} = \mathbf{U}_{3/4c} \\ \text{3D}: \mathbf{U}_\infty + \mathbf{U}_{ind,3D} = \mathbf{U}_{3/4c} \end{array} \right\} \Rightarrow \mathbf{U}_{rel} = \mathbf{U}_\infty + \mathbf{U}_{ind,3D} - \mathbf{U}_{ind,2D}, \tag{3}$$

where $\mathbf{U}_{3/4c}$ is the velocity at the three-quarter chord position, $\mathbf{U}_\infty$ is the freestream velocity, $\mathbf{U}_{ind,3D}$ is the velocity induced by the 3D vorticity system toward the control point ($P_j$), and $\mathbf{U}_{ind,2D}$ is the velocity induced by an infinite filament positioned at the one-quarter chord of the current section.

Induced velocities are calculated with the previous circulation distribution by using the Biot–Savart law, which relates the strength of a vortex filament to the magnitude and direction of the flow field that it induces. $\mathbf{U}_{ind,3D}$ is calculated as the sum of the velocities induced by all the sections, and $\mathbf{U}_{ind,2D}$ only takes into account each section's own circulation.

4. Calculate the effective angles of attack ($\alpha_{eff}$) with the direction of the relative velocities with respect to the airfoil and interpolate the lift coefficients ($C_l$) from the 2D airfoil polars.
5. Recalculate the bound circulation at each section with the obtained lift coefficients using the Kutta–Joukowski law, which relates the lift force ($L$) with the bound circulation, formulated as

$$\rho |\mathbf{U}_\infty \times \Gamma| = \frac{1}{2}\rho \left| \mathbf{U}_{rel} \times \hat{\mathbf{z}}_{airf} \right|^2 cC_l(\alpha_{eff}), \tag{4}$$

where $\rho$ is the air density, $c$ is the airfoil chord, and $\hat{z}_{airf}$ is the local unit vector along the airfoil's $z$-axis, in the direction of the lifting line.

6. Compare the updated circulation to the last iteration and check if it falls below the convergence criteria. If it does not, go back to step 3. For the next iteration, the circulation is calculated using an under-relaxation factor to stabilize the solution.
7. Recalculate the local angles of attack at the quarter-chord position using the converged circulation distribution.
8. Convert the local forces into the freestream velocity direction and derive the global lift and drag coefficients by integrating the forces along the span.

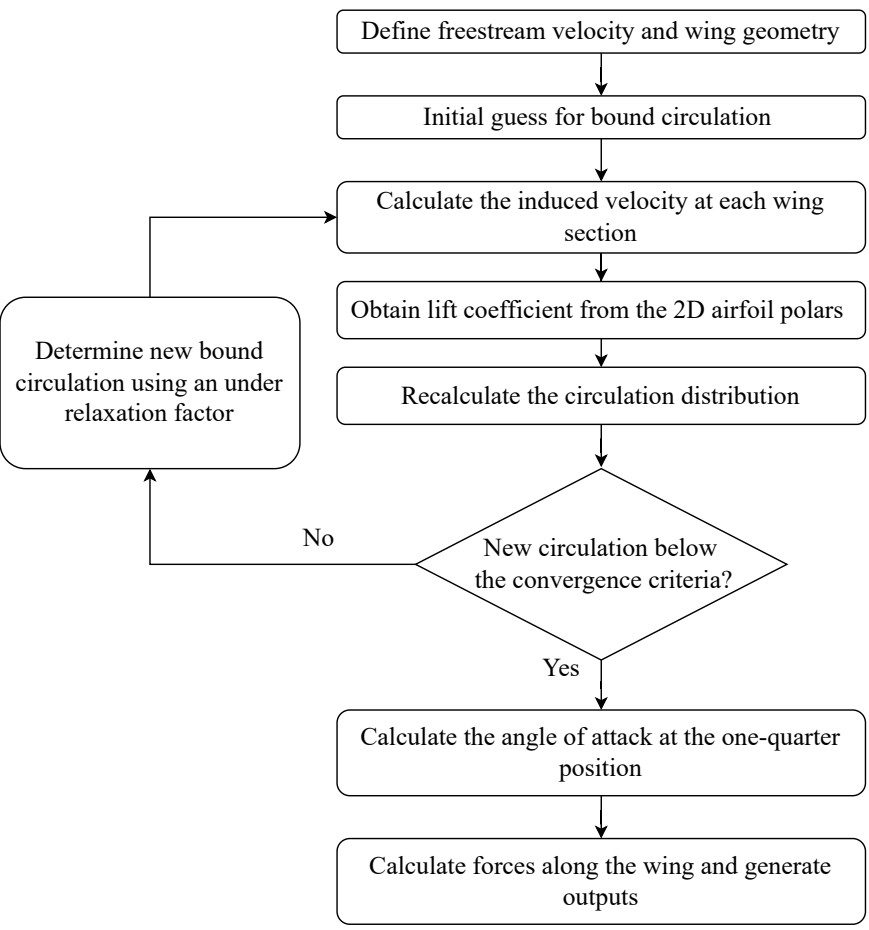

**Figure 4.** Flowchart of the vortex step method.

Figure 5 illustrates the discretization employed in the aerodynamic model, highlighting the need for a much higher level of refinement compared to the structural model.

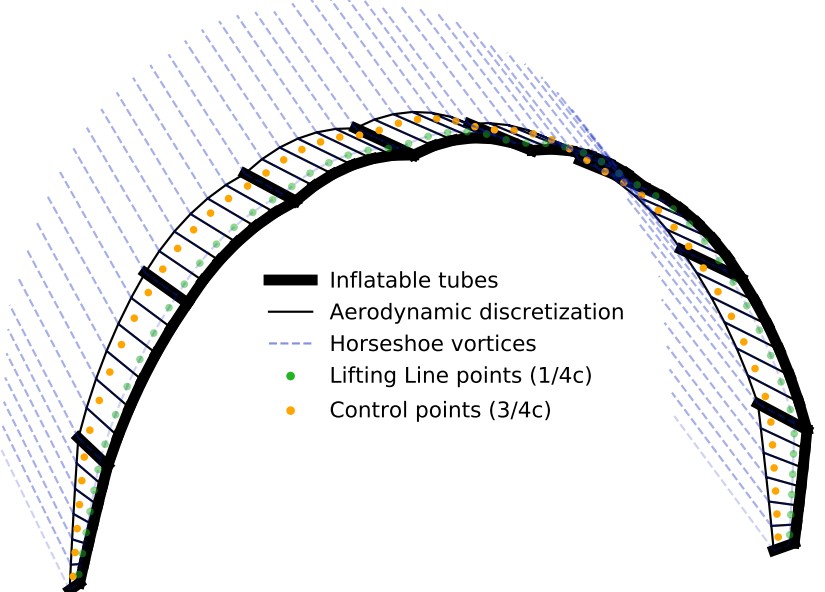

**Figure 5.** Aerodynamic model of the LEI V3 kite, illustrated for a coarse spanwise discretization of the wing [26].

### 2.3. Aero-Structural Coupling

The aero-structural coupling is based on the assumption that the deformation of the kite due to changes in the relative inflow or wing actuation can be described as a transition through steady equilibrium states. This quasi-steady assumption is justified because the time scale of inertial forces is substantially smaller than the time scale of the flow around the wing and the relevant larger-scale deformation phenomena that affect the kite behavior [7,32]. This assumption does imply that sub-scale deformation modes with high frequencies are neglected, such as TE flutter or seam rippling.

In the iterative coupling loop, the forces computed by the aerodynamic model are fed into the structural model, which computes a new deformation state, which in turn is fed back into the aerodynamic model. This loop is continued until a converged geometry of the kite and bridle-line system is determined (see Figure 6). However, for the solution to converge, it is necessary to add a constraint that fixes the particle system at one point in space and prevents it from moving freely. This is because the scope of this study is limited to the kite, not considering its flight along a trajectory in space. The constraint is obtained by fixing the bridle point (see Figure 1) in space. The approach can be regarded as a virtual wind tunnel, as described in [17,33], where the trim point was fixed in space. In the case of symmetric actuation of the wing, no resultant aerodynamic side force is generated, and the kite converges to a stable trim angle, which depends on the power state of the wing ($u_p$). On the other hand, when the actuation is asymmetric because the kite is steered, an aerodynamic side force is generated, and an additional constraint is needed to prevent a lateral motion of the kite.

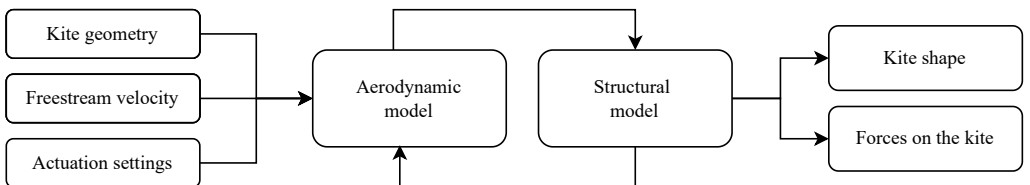

**Figure 6.** Flowchart of the coupled aero-structural model.

The interface between the two models is straightforward, as the aerodynamic model requires only the geometry of the nine connected quadrilaterals representing the wing segments. The quadrilateral mesh is then refined and adapted to the needs of the aerodynamic code, where a much finer mesh is required. The different mesh resolutions of the two models are illustrated in Figure 7 for a single wing segment between two struts. The red and blue lines represent the structural and aerodynamic meshes for this wing segment, respectively. The sectional aerodynamic forces ($F_{a_i}$) and moments ($M_{a_i}$) are applied at the one-quarter chord line, as represented by the blue dots. For the coupling of the two models, the aerodynamic forces need to be mapped from the relatively fine aerodynamic mesh of the wing segment to the four structural nodes represented by the red dots.

Commonly used, mesh-based interpolation methods converge very poorly because the fact that the two mesh resolutions strongly differ leads to a substantial information loss [34]. A pragmatic solution to this problem is to assume that the zone between the two meshes is part of a rigid body so that fundamental concepts can be adopted for the mechanical coupling. The proposed approach preserves the resulting forces and moments within each wing segment when mapping forces and moments from the coarse mesh to the fine mesh [26].

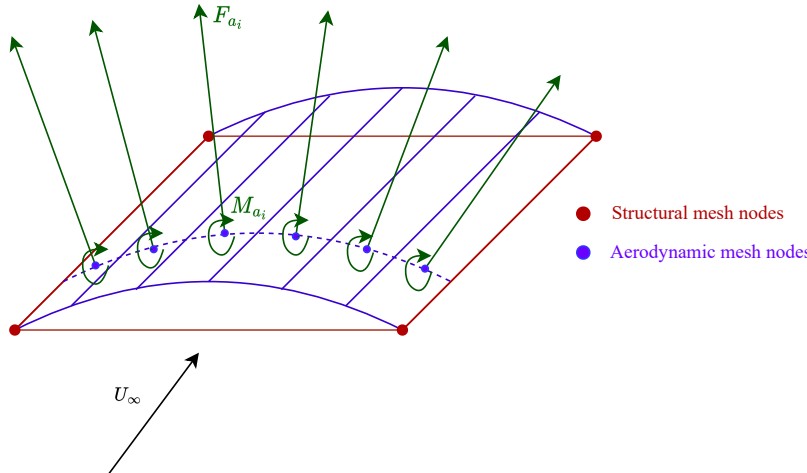

**Figure 7.** Structural and aerodynamic meshes with largely different spanwise resolutions [26]. While not illustrated in this schematic, the quadrilateral defined by the structural mesh nodes is not necessarily planar but skewed in the general case.

*2.4. Photogrammetry*

The photogrammetric analysis was based on video footage acquired during the flight operation of the kite, following the method described in [5]. The video camera was mounted on the KCU and directed toward the wing. This perspective allowed for a quantitative assessment of the planform of the wing as a function of the power setting. The tip-to-tip distances from the LE and TE were measured for the extreme actuation states, i.e., fully powered and fully depowered, assuming a linear transition in between those extremes. In addition, the individual TE distances were also measured, and their relative change was used in the simulations to account for canopy ballooning. The video stills were modified using an image processing tool to remove the optical distortion caused by the fish-eye lens. Other perspective distortions, caused, for instance, by the relative position of the KCU or by deformations, were corrected by taking the struts as a reference measure, assuming they do not vary in length. On the other hand, distortions induced during a turning maneuver were considered too large to be used quantitatively [25,26].

**3. Results**

Simulation results are presented separately for the deformation of the kite and the aerodynamics of the wing. Section 3.1 describes the computed steady-state deformation of the wing and bridle-line system. Section 3.2 describes the computed aerodynamic coefficients of the wing for different inflow conditions. The aerodynamic and structural models were also validated individually. In [26], the aerodynamic model was validated for different wings, including the effects of low aspect ratio, high anhedral angles, and sweep angles. In [25], the structural model was validated with a much more simplified aerodynamic model than the VSM presented in this paper.

*3.1. Kite Deformation*

In this section, the effect of the steering-line actuation on the shape of the wing is presented and discussed. The deformed wing shapes are shown for the extremes of each operating condition, i.e., fully depowered and fully powered in straight flight and with maximum steering input. For validation purposes, the evolution of the kite width, measured from the bridle-line attachment points at the tips of the LE (power lines) and TE (steering lines), is presented and compared with the photogrammetry results (see Figure 1).

### 3.1.1. CAD Geometry versus Powered Wing Shape

Firstly, the deformation of the powered kite is compared to the geometry of the CAD design of the kite. The CAD geometry is the initial input for the simulation and was previously used to analyze the aerodynamics of the kite using CFD [23,24]. Results deviating considerably from the available experimental data.

As shown in Figure 8, the shape of the powered kite is somewhat flatter than the CAD geometry due to the effect of aerodynamic forces. Furthermore, there is a noticeably greater LE width and a slightly smaller TE width in the CAD geometry, which means higher angles of attack in the outer sections, which hardly contribute to the lift of the wing, but they do contribute to the drag, decreasing the aerodynamic efficiency of the kite. This effect can be seen in the aerodynamic results presented in Section 3.2.

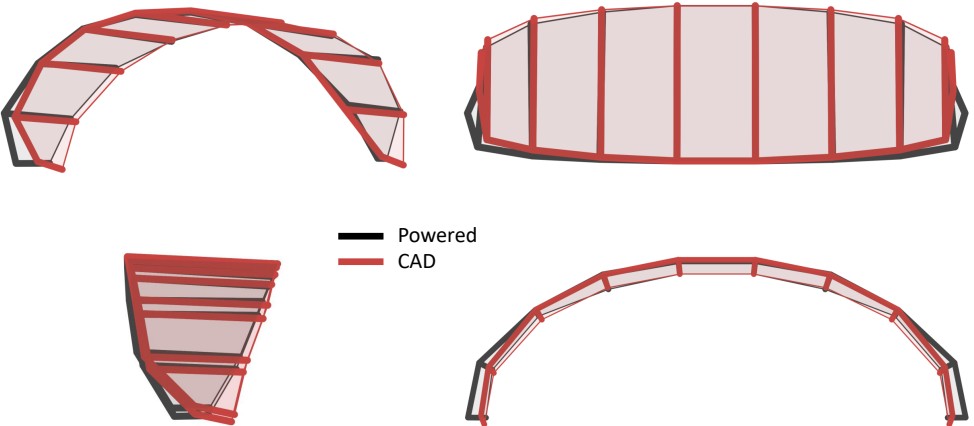

**Figure 8.** In black, the wing shape in a fully powered state ($u_p = 1$), and in red, the shape of the CAD model, displayed in an orthographic view (**top left**), a top view (**top right**), a side view (**bottom left**), and a front view (**bottom right**) [26].

### 3.1.2. Powered versus Depowered Wing Shape

In the following step, the symmetrical deformation caused by a length change in the depower tape was investigated. The length difference $\Delta l_d$ (see Figure 2 and [5]) between powered and depowered states was measured in two separate flight campaigns, yielding $\Delta l_d = 8\%$ and 13%, although it is unclear which of these values corresponds to the video-recorded flight. Therefore, both values were tested in the simulations.

Figure 9 shows a qualitative comparison of the computed powered and depowered kite shapes, for $\Delta l_d = 8\%$, using the canopy-billowing relations derived from photogrammetry. It can be seen that the tip distance decreases when depowering the kite. This decrease is more accentuated in the TE because it is connected to the steering lines which are affected by the depower tape.

This change in wing curvature induces two noteworthy changes in the aerodynamics of the kite. On the one hand, the angle of attack of the middle wing sections decreases by around 5°, according to the simulations, reducing the local aerodynamic loading and allowing a more efficient reel-in, which is desired when depowering the kite. On the other hand, however, the angles of attack of the outer sections increase due to the more significant decrease in TE width, increasing the drag of the kite without contributing significantly to the lift.

For quantitative validation of this effect, the geometric wing-tip distances of the powered and depowered kite were evaluated by photogrammetry, using available video stills. Figure 10 shows the evolution of the width with the powering input, with $u_p = 1$ describing the fully powered state. One can observe that the modeling of the canopy-billowing effect considerably affects the shape of the kite. Accounting for the TE segment length variation decreases the tip distances by a further 2–6%. Interestingly, by accounting for the canopy billowing, a depower tape difference of $\Delta l_d = 8\%$ or 13% results in the same

widths in the depowered state. Therefore, an increase in the depower tape past 8% does not affect the shape of the kite significantly. Thus, the simulation suggests that reeling out the depower tape further during flight does not make a difference in further depowering the kite.

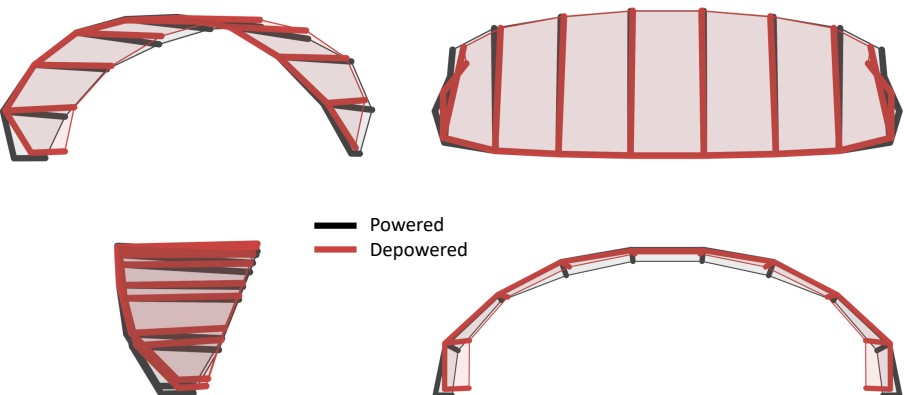

**Figure 9.** In black, the wing shape is in a fully powered state ($u_p = 1$), and in red, the wing shape is in a fully depowered state ($u_p = 0$), for $\Delta l_d = 8\%$, displayed in an orthographic view (**top left**), a top view (**top right**), a side view (**bottom left**), and a front view (**bottom right**) [26].

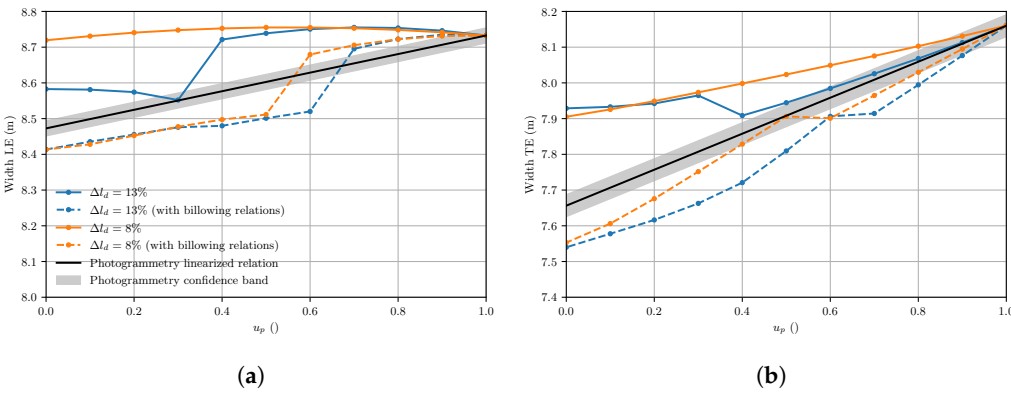

(**a**)                                                                                              (**b**)

**Figure 10.** Evolution of the LE widths (**a**) and TE widths (**b**) as a function of the power setting $u_p$.

The photogrammetry provided only the extreme values of the tip distances—i.e., the linear dependency being in between $u_p = 1$ and 0 is a pure assumption. In fact, the simulations showed more complex behavior with an abrupt change that corresponds to the state in which the steering tape and the bridle line connecting it to the TE tip are parallel. This is illustrated in Figure 11.

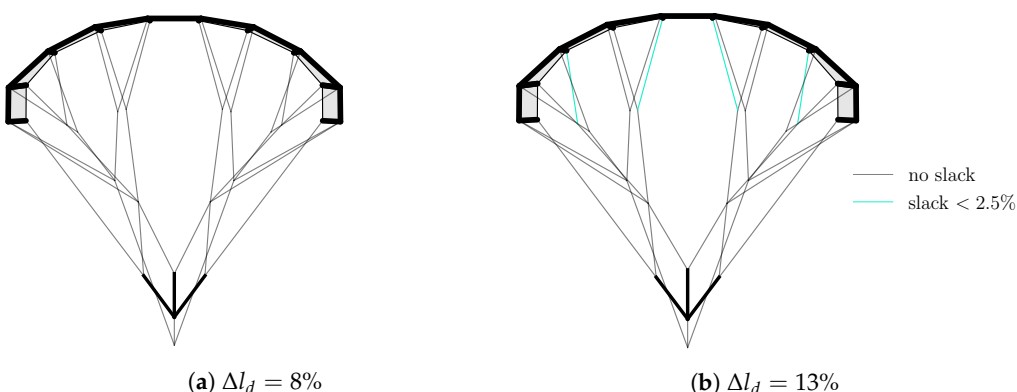

(**a**) $\Delta l_d = 8\%$                                                                    (**b**) $\Delta l_d = 13\%$

**Figure 11.** Frontview of the depowered kite ($u_p = 0$) indicating the tensioning state of the bridle-line system [26].

Since those two lines cannot move relative to each other, an increase in power-tape length results in an increase in TE curvature, pulling the LE tips inwards, which causes an abrupt change in how the wing shape evolves with the power setting.

For further validation with the available video footage, the movements of the pulleys and the steering tape knots were also tracked. When going from powered to depowered, both the simulations and the video footage show that the pulleys move inwards, whereas the knots move outwards. Finally, the last qualitative validation was performed by comparing the slacking bridle lines. In the powered state, which is relatively similar to the nominal wing design, no slack was observed, neither in the simulations nor in the video footage. However, four slacking lines could be observed in the video footage of the depowered kite, which only appeared for simulations exceeding $\Delta l_d = 8\%$, while the geometry of the kite remained relatively constant. Furthermore, two slacking lines in the simulations match with the slacking lines in the video footage, and the two other slacking lines are attached only to the same wing section as the lines in the video footage [26].

As a concluding remark, although the results are promising, there are many lines in the bridle-line system whose initial length is unknown. In addition, it was observed during the simulations that a slight change in the initial geometry of the bridle-line system affects the slacking behavior of the system. Finally, although efforts have been made to correct the distortions in the photogrammetry measurements, there are still many factors causing uncertainty in the measurements [26]. Thus, these results should help identify trends rather than be used for quantitative validation.

### 3.1.3. Wing Shape for Powered Straight Flight versus Turning Maneuvers

The last actuation case presented is the asymmetric deformation caused by the steering of the kite. The mechanism of steering of an LEI kite is a combined effect of several phenomena and explained in greater detail in [7,10,35]. The aero-structural response of the wing to lengthening one steering tape and shortening the other is visualized in Figure 12 and compared to the powered state in straight flight. Note that the following discussion is from the perspective of an observer flying with the kite and looking in the direction of flight. That means in a front view of the kite, the right wing tip (from this kite-attached observers' perspective) appears on the left side from the perspective of the reader. In the discussed case, the kite is making a right turn (from the observers' perspective) or maneuvering to the left (from the reader's perspective). This maneuver is performed by increasing the angles of attack on the right side of the wing relative to the left side, which induces a combined roll and yaw moment that causes the wing to turn.

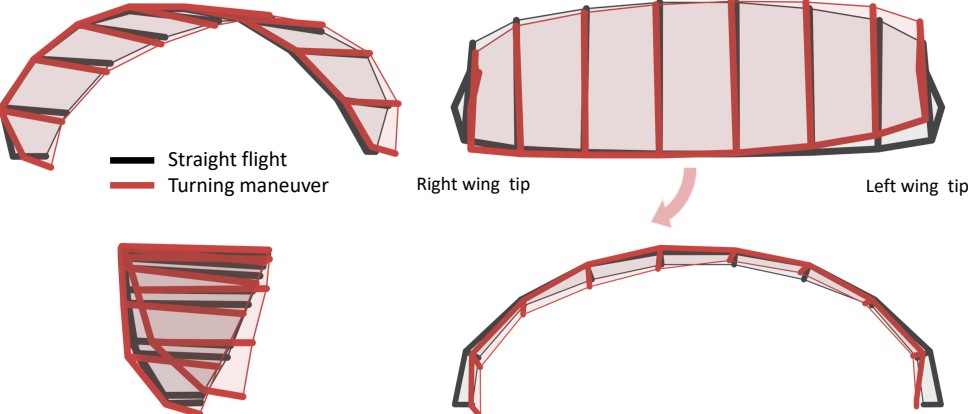

**Figure 12.** In black, the wing shape for no steering input ($u_s = 0$), and in red, the wing shape for the maximum steering input ($u_s = 0.4$), displayed in an orthographic view (**top left**), a top view (**top right**), a side view (**bottom left**) and a front view (**bottom right**). Adapted from [26].

It can be seen qualitatively that the computed deformation behavior of the kite agrees well with the video footage, where the right wing tip moves forward (yaw moment) and downward (roll moment). Furthermore, the slacking lines in the simulation (see Figure 13) are compared to the footage, and in this case, the simulations resulted in two extra slacking lines that do not appear in the video footage. These are the lines connecting the steering lines to the power lines, and their appearance is attributed to the extra constraint set in the steering states that restraints the kite from moving laterally. On the other hand, the last slacking line coincides with the in-flight video recordings of a turning maneuver [26].

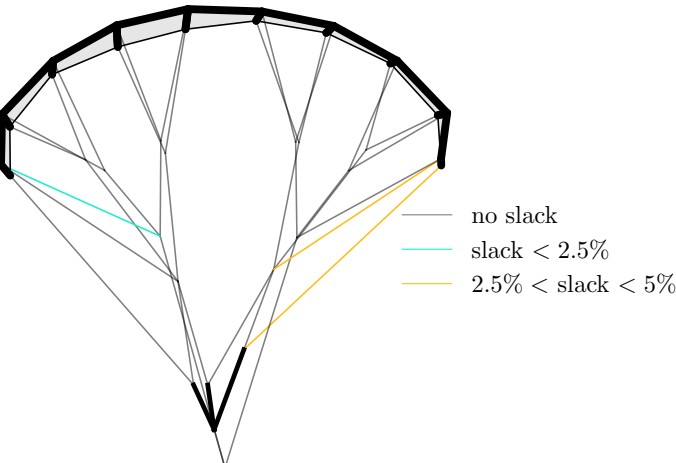

**Figure 13.** Front view of the kite for maximum steering input ($u_s = 0.4$), indicating the tensioning state of the bridle-line system [26].

### 3.2. Kite Aerodynamics

This section focuses on the aerodynamics of the wing. For each actuation state, the aero-structural deformation of the kite is computed first, and then the aerodynamics of the resulting shape are studied in detail using the VSM for various relative flow conditions. For all presented cases, the angle of attack is defined with the center section of the wing.

#### 3.2.1. Computed Aerodynamic Properties

Figure 14 shows the drag and lift curves for three extreme deformation states together with the polars of the CAD model of the kite using VSM and RANS. The polars of the CAD model serve as a cross-reference for comparing the VSM to the higher-fidelity method, and the polars with a deformed state indicate how the deformation affects the aerodynamics of the kite.

For the CAD geometry, the values of $C_L$ and $C_D$ are close to those of the CFD simulations in the linear region of the lift, differing by no more than 0.15 and 0.01, respectively. However, a difference in the lift slope can be seen, which is attributed to the 2D polars obtained with Breukel's model [7], which also caused a difference in the lift slope compared to 2D polars obtained with CFD. Finally, the VSM predicted the stall at a lower angle and with a slightly lower $C_{L_{max}}$. Potential flow-based methods generally tend to fail when exceeding the stall angle, and this is no exception. As the implemented VSM makes use of 2D viscous polars, which change slope when passing the stall angle, several numerical solutions exist for the circulation distribution problem near the stall angle—some of them being non-physical. Consequently, solutions that are physically unfeasible may appear, usually resulting in an overestimation of lift and an underestimation of drag [26].

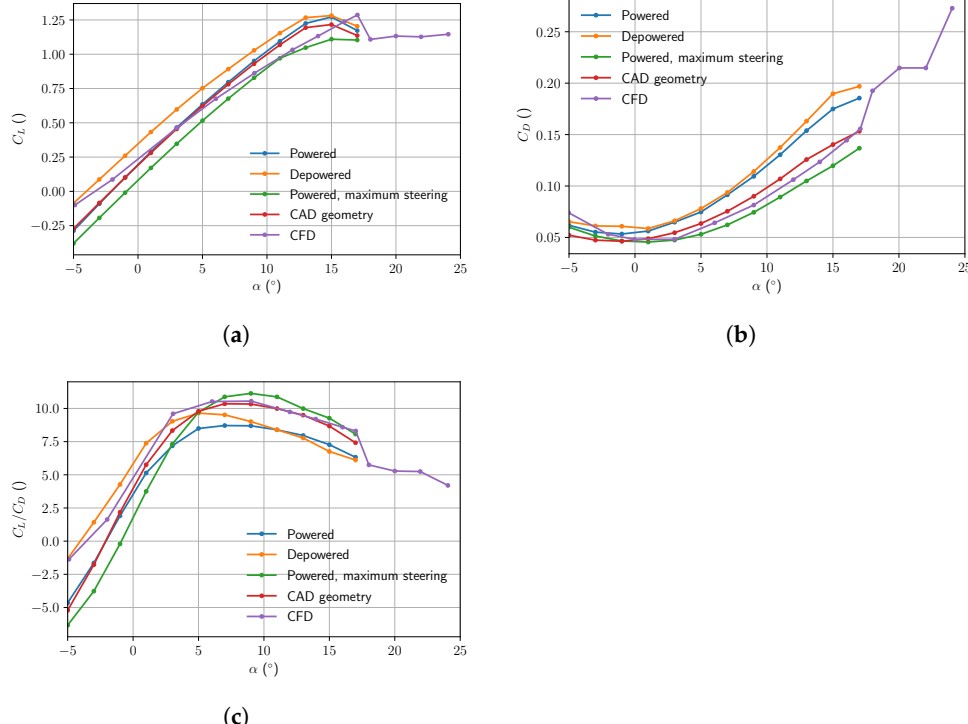

**Figure 14.** Lift coefficient (**a**), drag coefficient (**b**), and lift-to-drag ratio (**c**) as functions of the angle of attack for different kite geometries.

The lift curve of the powered kite is almost identical to the curve of the CAD model, with the only difference being that the lift is slightly higher for larger angles of attack. However, the drag is higher for all angles of attack due to increases in the relative angles of attack of the outer sections of the wing. These sections hardly contribute to the lift component in the tether direction but contribute to the drag and consequently also lower the aerodynamic efficiency of the kite.

For the depowered state, the change in the wing shape results in higher lift for the same angle of attack of the middle section, while the drag remains at very similar values to those in the powered state. Therefore, depowering the kite results in a more aerodynamically efficient shape, and its maximum efficiency is reached at a lower angle of attack, in the range where the depowered kite usually operates [36].

Lastly, focusing on the asymmetrically deformed kite during steering, lift and drag both decrease with respect to the powered state without steering input. This decrease in lift can also be observed experimentally. However, the drag does the opposite: it increases. This discrepancy with the experimental data can be explained in several ways. Firstly, the aerodynamic sideslip angle is generally very high during turns, whereas in these idealized virtual wind-tunnel simulations, the sideslip angle is kept at zero by constraining the lateral motion component of the wing. As a result of the turning motion at a relatively small radius, the relative flow velocities along the span will vary, which affects the aerodynamic load distribution on the kite, and consequently, the shape.

### 3.2.2. Comparison with Experimental Results

In this section, the quality of the PSM-VSM model is assessed on the basis of two experimental studies [5,36]. Both studies used engineering models and flight-test data of the LEI V3 kite to identify its aerodynamic properties. The simpler model used in [5] only yielded the resultant lift, and drag force components and the identification were based on a limited set of five available pumping cycles. The more advanced approach presented in [36] also provided the aerodynamic moments using a three-plate model of the wing. With 182 pumping cycles, the statistical quality of the experimental dataset was substantially higher.

In addition, corrections were added to [5] to account for the inertial properties of the KCU, the wing, and the weight and drag of the tether.

The angle of attack was measured with a pitot tube and corrected using a geometric expression that links the length of the depower tape to the variation in wing pitch angle caused by the deformation of the wing. However, the pitch-angle difference between powered and depowered states calculated with these simple geometric expressions differed from the values found with the aero-structural model. Moreover, uncertainties were observed due to the precision of the GPS readings used to follow the trajectory. A second source of uncertainties was introduced by the asymmetric mounting of the pitot tube in the setup used in [36]. As the apparent wind speed was measured in one fork of the power lines, the variation in the inflow speed along the wing span during sharp turning maneuvers introduced a certain deviation depending on the turning direction. For these reasons, it was not possible to rigorously assess the accuracy of the experimental results. They were consequently used to identify trends rather than for quantitative validation.

With that in mind, the comparison is presented in Figure 15, along with the CFD simulation of the undeformed kite. The results were obtained by defining a range of inflow and actuation conditions for each operation mode, i.e., straight-powered, turning-powered, and depowered. These values are presented in Table 1 and are based on several experimental studies of the LEI V3 kite [5,36,37].

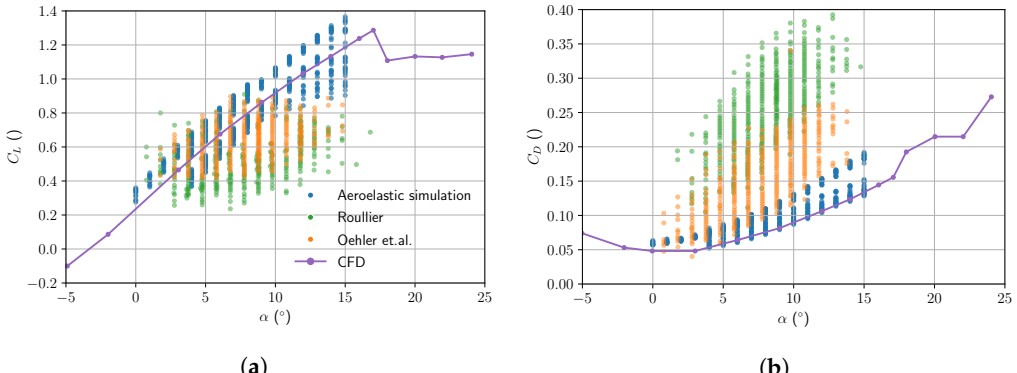

(**a**)                                                                                  (**b**)

**Figure 15.** Lift coefficient (**a**) and drag coefficient (**b**) of the kite, obtained with the experimentally based flow and steering inputs, and compared to CFD and experimental studies in [5,36].

**Table 1.** Steering and relative flow inputs representing the different flight maneuvers of a pumping cycle.

|  | Angle of Attack $\alpha$ (°) | Angle of Sideslip $\beta$ (°) | Power Setting $u_p$ | Steering Setting $u_s$ |
|---|---|---|---|---|
| Straight-powered | 4–14 | 2–8 | 0.7–1 | 0 |
| Depowered | 0–8 | 0–4 | 0–0.6 | 0 |
| Turn-powered | 9–15 | 2–10 | 1 | 0.1–0.4 |

The values of the lift coefficient are within the range of the experimental results up to an angle of attack of about 10 degrees. Above this value, the lift is overestimated. As for the drag coefficient, there is a tendency to increase with respect to the CFD simulations, as in the experimental results, although compared to the experimental results, the values are still considerably smaller, especially for large angles of attack.

One of the reasons for this discrepancy with the experimental data comes from the main limitation of the VSM, which is the inability to deal with stall. This limitation is particularly problematic in an LEI kite due to its curvature, since with a sideslip inflow component, part of the wing stalls earlier than it would for a straight wing. As discussed above, this limitation results in an overestimation of lift and an underestimation of drag, which is the general trend shown in Figure 15. This divergence is especially prominent at large angles of attack due to the wing entering stall earlier.

## 4. Discussion and Conclusions

An aero-structural model of LEI kites was presented as the computational core component of a tool for the design phase of airborne wind-energy systems. As such, the model has to be fast and at the same time accurate, especially when predicting kite deformations that substantially affect the aerodynamic performance of the kite. The developed Python implementation of the PSM-VSM model requires around 5 min on a common laptop to compute a steady deformation state from the CAD geometry as the initial value. This run time varies depending on the applied actuation. On the other hand, in each time step of the dynamic simulation, the run time for the aerodynamic and structural models is 1–2 s each. Consequently, if this model were to be included in a dynamic flight simulation, each time step would use the result of the previous time step as an initial value, which would drastically reduce the computational effort per time step.

As for the accuracy of the model, it can reproduce the deformation modes realistically, resulting in aerodynamic results that are closer to experimental data than previous analyses with higher-fidelity aerodynamic models. That said, several aspects should be investigated in future work to make the model more reliable, faster, and more robust.

Firstly, it should be studied how to include realistic properties of the spring-damper elements representing the bridle lines, allowing real-time simulations to be carried out. Similarly, how to represent the deformation of the canopy and inflatable tubes should be investigated, so that it is not necessary to rely on experimental canopy-billowing relations. In addition, the weights and drag of the KCU, wing, and bridle lines could be included to make the model more realistic.

Secondly, a different constraint should be found for the simulations to converge on a steady state. Currently, the bridle point is fixed in all directions for cases with symmetric actuation, and for asymmetric actuation, the lateral movement of the wing is also constrained. For the symmetrical deformation cases, this means that only the deformation at the stabilization angle of the kite is calculated, whereas for the asymmetrical cases, an unrealistic constraint is used, which results in extra slacking lines. To solve this problem, simulations of the kite trajectory would have to be done and either include the complete system with the tether or create a constraint so that the kite can move in one plane, thereby avoiding imposing a constraint on the wing.

Thirdly, one of the major weaknesses found stems from the inability of the VSM to correctly predict stall, which appears earlier when flying turning maneuvers that introduce a sideslip angle than a in straight flight. Consequently, it is recommended that efforts be devoted to improving the aerodynamic model so that it can accurately predict post-stall angles of attack. Continuing with the aerodynamic model, the VSM is coupled to a 2D model that generates the viscous polars of LEI airfoils. This model was compared with experimentally determined aerodynamic properties and with CFD results, although it is not certain how reliable the model is because all three differ [26]. Additionally, the moment coefficient was shown to have very high values. Furthermore, the correlation model does not account for variations in Reynolds number or maximum camber position. For these reasons, it is recommended that a more refined model be developed to generate 2D polars of LEI airfoils.

With all these aspects of improvement, there is still one issue to be addressed in order to test the accuracy of the model quantitatively, which is obtaining more reliable experimental validation data. To that end, an LEI kite should be flown with all the bridle-line lengths known and the kite fully characterized. The measurement sample size should be increased, and the videos for photogrammetry analysis should be captured at the same time as the flow and actuation conditions so that both datasets can be easily correlated. Additionally, to eliminate distortions, the photogrammetry analysis might be performed with two cameras, ideally without the wide-angle lens. Alternatively, other methods could be used to measure deformations, such as the use of inertial measurement units or a 3D scanning technique.

In summary, a pioneering aero-structural model was presented which promises to be a useful tool for the optimization of LEI kites. However, due to the complexity of the problem, there are still many aspects that could be refined to make this model more accurate and faster, so further research is encouraged.

**Author Contributions:** Conceptualization, O.C., M.G. and R.S.; methodology, O.C., M.G. and R.S.; software, O.C.; validation, O.C.; formal analysis, O.C. and M.G.; writing—original draft preparation, O.C.; writing—review and editing, M.G. and R.S.; visualization, O.C.; supervision, M.G. and R.S. All authors have read and agreed to the published version of the manuscript.

**Funding:** This research received no external funding.

**Data Availability Statement:** A Python implementation of the vortex step method described in this paper is available from the project repository: https://github.com/awegroup/Vortex-Step-Method, accessed on 20 March 2023.

**Conflicts of Interest:** The authors declare no conflict of interest.

## Abbreviations

The following abbreviations are used in this manuscript:

| | |
|---|---|
| 2D | Two-dimensional |
| 3D | Three-dimensional |
| AWE | Airborne wind energy |
| CAD | Computer-aided design |
| CFD | Computational fluid dynamics |
| FE | Finite element |
| FEM | Finite element method |
| FSI | Fluid–structure interaction |
| KCU | Kite control unit |
| LE | Leading edge |
| LEI | Leading-edge inflatable |
| LLM | Lifting line method |
| PSM | Particle system model |
| TE | Trailing edge |
| RANS | Reynolds-averaged Navier–Stokes |
| VSM | Vortex step method |

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
