# Peer review of "Fast Aero-Structural Model of a Leading-Edge Inflatable Kite"

_energies, doi:10.3390/en16073061_

Round 1

Reviewer 1 Report

In this paper the authors have presented a fast aeroelastic model for wind kite design, by coupling a particle system based structual model and an aerodynamic model. The model is proven to be accurate by validating with several wing geometries and experimental data. Overall this paper is of great quality and can contribute to the airbone wind energy community. I recommend publishing this paper. Only one minor suggestion:

1. Line 150: It is better to have a paragraph to provide more detail the RANS model used in this study, although the authors have referred to a PhD thesis here.

Author Response

Dear reviewer,

Thank you for taking the time to review our manuscript and providing feedback. Please find below a response to your remarks.

In this paper the authors have presented a fast aeroelastic model for wind kite design, by coupling a particle system based structual model and an aerodynamic model. The model is proven to be accurate by validating with several wing geometries and experimental data. Overall this paper is of great quality and can contribute to the airbone wind energy community. I recommend publishing this paper. Only one minor suggestion:

%%%%
Point 1: Line 150: It is better to have a paragraph to provide more detail the RANS model used in this study, although the authors have referred to a PhD thesis here.

Author response: We agree. Since this model is integrated into the aeroelastic code, it makes sense that the reader can quickly see what type of model has been used to obtain the aerodynamic coefficients.

Best regards,
The authors

Reviewer 2 Report

This paper presents a fast aeroelastic model of leading-edge inflatable kites for the design phase of airborne wind energy systems. The fluid-structure interaction solver couples two fast and modular models, a particle system model to capture the deformation of the wing and bridle line system and a 3D nonlinear vortex step method coupled with viscous 2D airfoil polars to describe aerodynamics. The flow solver was validated with several wing geometries and proved to be accurate and computationally inexpensive for pre-stall angles of attack. The coupled aeroelastic model was validated using experimental data showing good agreement in the deformations and aerodynamic forces. Therefore, the speed and accuracy of this model make it an excellent foundation for a kite design tool.

1.                   In the Introduction, There is a syntax error in the first sentence. The noun phrase Wind seems to be missing a determiner.

2.                   Figure 2 is not clear enough to show the Y-splits of the bridle close to the tubes.

3.                   In the Computational Approach, in line 136, there should be an “a” before “three-quarter” and in line 140, the singular noun magnitude follows a number other than one, it should change the noun to the plural.

4.                   In the figure 5what does the “  mean?

5.                   Fast Aeroelastic Model of a Leading-Edge Inflatable Kite is inseparable from high-performance materials and advanced manufacturing technology. The following latest references may be considered for citation in this paper.

6.                   (i)(2022) Programmable Thermo-Responsive Self-Morphing Structures Design and Performance.?Materials,?15(24), 8775.

7.                   (ii)(2022) Quality Quantification and Control via Novel Self-Growing Process-Quality Model of Parts Fabricated by LPBF Process.?Materials,?15(23), 8520.

8.                   (iii)(2022) Quality Prediction and Control in Wire Arc Additive Manufacturing via Novel Machine Learning Framework. Micromachines, 13(1), 137.

9.                   (iv)(2022) Porosity management and control in powder bed fusion process through process-quality interactions, CIRP Journal of Manufacturing Science and Technology,  38:120-128.

Author Response

Dear reviewer,

Thank you for taking the time to review our manuscript and providing feedback. Please find below a response to your remarks.

This paper presents a fast aeroelastic model of leading-edge inflatable kites for the design phase of airborne wind energy systems. The fluid-structure interaction solver couples two fast and modular models, a particle system model to capture the deformation of the wing and bridle line system and a 3D nonlinear vortex step method coupled with viscous 2D airfoil polars to describe aerodynamics. The flow solver was validated with several wing geometries and proved to be accurate and computationally inexpensive for pre-stall angles of attack. The coupled aeroelastic model was validated using experimental data showing good agreement in the deformations and aerodynamic forces. Therefore, the speed and accuracy of this model make it an excellent foundation for a kite design tool.

%%%%
Point 1: In the Introduction, There is a syntax error in the first sentence. The noun phrase Wind seems to be missing a determiner.

Author response: We think it is not necessary in this context. The sentence is referring to "wind" as a general concept or substance, rather than a specific instance, so using "the" or another determiner would not be necessary.

Point 2: Figure 2 is not clear enough to show the Y-splits of the bridle close to the tubes.

Author response: These photos are the best that we could find from a large selection of photos that detail this specific kite in flight. In our opinion, especially the right photo in Figure 2 clearly shows the Y-splits close to the struts, which, together with the CAD drawing in Figure 1, should provide sufficient clarity.

Point 3: In the Computational Approach, in line 136, there should be an “a” before “three-quarter” and in line 140, the singular noun magnitude follows a number other than one, it should change the noun to the plural.

Author response: Agreed and fixed.

Point 4: In the figure 5 what does the “”  mean?

Author response: We do not understand what "" refers to. We added an explanation of U_infinity, if that is what you meant.

Point 5: Fast Aeroelastic Model of a Leading-Edge Inflatable Kite is inseparable from high-performance materials and advanced manufacturing technology. The following latest references may be considered for citation in this paper.

Author response:  We agree that performance materials and advanced manufacturing technology are intrinsicly connected to the design of kites, but we believe that the references that the reviewer mentions are not relevant to our manuscript.

Point 6: (i)(2022) Programmable Thermo-Responsive Self-Morphing Structures Design and Performance.?Materials,?15(24), 8775.

Author response: ?

Point 7: (ii)(2022) Quality Quantification and Control via Novel Self-Growing Process-Quality Model of Parts Fabricated by LPBF Process.?Materials,?15(23), 8520.

Author response: ?

Point 8: (iii)(2022) Quality Prediction and Control in Wire Arc Additive Manufacturing via Novel Machine Learning Framework. Micromachines, 13(1), 137.

Author response: ?

Point 9: (iv)(2022) Porosity management and control in powder bed fusion process through process-quality interactions, CIRP Journal of Manufacturing Science and Technology,  38:120-128.

Author response: ?

Best regards,
The authors

Reviewer 3 Report

Please see comments on attached PDF document. The paper is well presented but the impact of the paper can be increased by addressing the comments provided. 

Author Response

Dear reviewer,

Thank you for taking the time to review our manuscript and providing feedback. Please find below a response to your remarks.

General comments:
Interesting paper worth of publication subject to the following corrections and enhancements.

Introduction

%%%%
* Justification? What does fast mean? Fast and reasonably accurate?

Author response: We agree that it sounds ambiguous. What we wanted to convey in this paragraph is that the models currently available are either too simple, which does not allow the structural and aerodynamic phenomena to be reproduced, or too complex, which makes iterative simulation, necessary for the design and optimization of these systems, too costly.

Comments on Figures

* Figure 1. The are many angle variables, especially on the side view of the figure, that need to be introduced in the text, please explain them when referring to the figure.

Author response: A clearer explanation is included and the angles that are not used in the paper have been removed.

* Figure 1. Please also refer on the figure which one is the leading an trailing edge of the kite. (Side view in particular). It is also not clear where in the diagram is the KCU. Is it the gray box down at the bottom? Please specify on the figure.

Author response: Agreed, we hadn't thought about the perspective of someone unfamiliar with these devices. This is now addressed.

* Figure 3. It would be useful to see on the Figures labels (1-9) for the 9 segments of the wing. The elements that are sued for the TE of the wing, whose contraction effect is modelled need to be highlighted on the figure.

Author response: We do not think it is necessary to label the 9 segments of the kite because we do not to refer to each of them individually in any of the arguments made in the paper.
Nevertheless, we have changed the figure to show where the trailing edge is and to make it more intuitive to understand the design of the kite.

Structural model

*The specific of properties for the spring and damper systems of each of the modelled lines could be specified in a table. This would be useful to readers working with similar models. In particular, the difference between TE elements that collapse, and non-collapsible elements would be clearer.

Author response: We do not find it necessary to specify them in a table because all of them have the same values. The parameters are now included in the text, along with the force equations.
The trailing edge billowing is modelled with experimental data that allows to relate the variation in the TE length to the power state (aerodynamic loading) of the kite.
This solution is not definitive and future work will have to consider how to include more realistic properties to model this effect without the need for experimental data. We are continuing to work on this.

* The assumption of minimal local FSI effects is similar to downwind sails at optimum trim. However, wind gusts could have an important effect on the shape of the kite.
Please comment on the limitations of the structural model.

Author response:

While there are some similarities between the canopy of a kite and the sail of a boat, there are also several clear and important differences.

Firstly, the canopy of a soft kite AWE device is much more constrained than a downwind sail due to the presence of inflatable tubes, which restrain the movement of the fabric more effectively than a sail. This allows the kite to maintain a stable and predictable shape, even in turbulent winds, reducing the risk of sudden and unexpected movements.

Secondly, soft kite AWE devices are designed to operate at much higher speeds than sailing boats, with typical flight speeds ranging between 10 to 20 meters per second, while a sailboat's top speed is around 4 meters per second. This means that they can fly at speeds that are much faster than the wind speeds, thus being less affected by sudden wind gusts.

Taken together, these factors mean that wind gusts have much less impact on a soft kite AWE device than on a sailing boat. The constrained canopy of the kite, combined with its ability to maintain a stable and predictable flight path, allows it to withstand sudden changes in wind conditions with greater ease.

Finally, this model is created with the aim of assessing the loads during nominal operation caused by different steering inputs. Although it would be interesting to assess the effect of wind gusts, this is not considered at this stage and the limitations are still not known.

* What software is used for the structural model? How is it implemented. Please specify for interested readers.

Author response: The model has been fully programmed in Python, both the structural and the aerodynamic models. This is mentioned in the discussion. To have more insights on the structural model, please refer to the MSc thesis of Jelle Poland [25], who is also publishing a parallel paper in this special issue.

Aerodynamic model

* The authors state that the magnitude of the circulation of each vortex ring is determined by using a control point at three quarter chord length. Then that the direction of the forces are determined using the quarter chord position. This is not entirely clear to me. The output of a panel method is indeed the magnitude of the circulations and the forces (lift and drag) on the wing. The direction of lift and drag are determined by the direction of the free stream. Therefore, the consideration of the quarter chord position to determine the direction of the forces needs to be better
explained by the authors.

Author response: We understand that there can be confusion here. The magnitude and direction of the forces are determined by the local angle of attack, which takes into account the velocities induced by the vorticity system. This is done in two different control points. Then, once calculated, they are transformed to the direction of the freestream velocity, which is the conventional way to define the aerodynamic forces. Therefore, the results are in that direction. A incise on that matter has been made in the text.

* LN143 the three-quarter angle of attack > the three-quarter chord position

Author response: Yes. Fixed.

* What are the kinematics used in the aerodynamic model? Can the authors provide
the flow boundary condition equation for interested readers, to understand the
model better?

Author response: These doubts are adressed with a flowchart explaining how the model works and the relevant equations that are used to reslove the problem. For a more detailed explanation of the boundary conditions the author refers to [26-29].

* The coupling of the VSM to the RANS simulations needs to be expanded. At which
point of the VSM method is the RANS input required? Please explain with a block
diagram, where you indicate the steps followed in the VSM and how the RANS
computations are used in the diagram.

Author response: This is now adressed with a block diagram, followed by an explanation of the iterative process used to obtain the aerodynamic results.

Aeroelastic coupling

* Fig 5. I think this figure is more illustrative of the aerodynamic model,
because it shows clearly some of the concepts used in the aerodynamic
model. Is the drag force computed? Please add to figure. Then, a new figure
showing block diagrams and the coupling between aerodynamic and
structural model, inputs and outputs could be beneficial.

Author response: We believe that the clarifications on the aerodynamic model are sufficient. The force Fa is the sum of lift and drag, and it is drawn to clarify where are they applied, not their direction. A flowchart with the inputs and outputs has been added.

Results

*Fig 6. The differences referred in the text between CAD and real kite are barely discernible in the Figure. Please add annotations to figure to show, the flatter shape, width of LE and TE, etc. Please also add legend box showing red is real kite and black is CAD. Are the 4 views necessary? Can the authors explain the shape differences with less figures and making them bigger?

Author response: Although not all the figures are used to describe the shape differences in this case, they are necessary for other case comparisons. The four figures are kept for two reasons: consistency on the deformation results and because it allows the reader to have a clearer idea of the 3D shape of the kite.

*LN247 was corresponds > corresponds

Author response: Yes. Fixed.

*Fig 8. Width of LE and TE. For readers that are not familiar with this terminology, it
would be useful to add an annotation to one of the figures (maybe figure 2), where
the widths are shown. Please be consistent in the terminology used in the caption of
the figure, refer to either length or width of LE and TE.

Author response: We added some notes on this in the first paragraph of section 3.1 and also in Figure 1. The terminology is consistent now.

*Explain briefly how is the CAD lift, drag and l/d curves computed. This is not clear in
the manuscript.

Author response: Explained in the aerodynamic model.

*LN268-269 it does actually not make sense > it does not make a difference

Author response: We agree, this is more accurate.

*Fig 9. Are the magenta and orange legends needed? If not, please remove them.

Author response: Not needed indeed. Done.

*Fig 10. It would also be useful to see a comparison of the slacked lines or
deformation of the kite from photogrammetry to those presented in these figures. I
think that Figure 10 and 11 can be merged into 1 figure, by only showing the top
view of Figure 10. Because this the view which highlights the steering manoeuvre.

Author response: Again, we believe that the 4 figures help the reader get the 3D shape better. The slacking lines are explained in the text. The photogrammetry photos showing the slacking lines can be found in the master thesis of this paper [26], which is cited in the same paragraph.

*Please explain how is the pitot tube measurement used to estimate the AoA. Provide
equation. How many pitot tubes are needed? This type of information is useful for
other studies non related to kites as well. It is hard to imagine that only one pitot
tube is used for such a complex geometry.

Author response: The experimental studies that collected the data to which the model is compared are referenced in the paper [5,36]. There, one can find a detailed explanation of the experiment setup. Since this paper only uses their results, it is not deemed relevant to clarify the setup nor the processing method of the experiments, which can be found in the references.

*Fig 13. Please provide labelled envelope lines around the simulations data to know
which points correspond to straight-powered, depowered, turn-powered.

Author response: We intentionally did not do this, as the purpose of Figure 15 is to compare the simulation results with experimental data, which are not divided into these categories. This approach allows for a more direct comparison between the simulation and experimental results, providing a clearer understanding of the system's performance. For a more detailed analysis of the aerodynamic performance of each actuation state the previous figure is provided.

*Fig 13. It is not clear where the VSM model start to fail. It seems reasonably accurate
in the range of data presented.

Author response: This is explained in the second paragraph of section 3.2.1. If the reader wants to see a plot with some of this nonphysical behaviours, [26] is referenced.

Discussion and results

* If possible, provide VSM python code as supplementary material and commented.

Author response: The Python code of both the PSM and the VSM is available from a public repository on GitHub via the link provided in the Data Availability Statement at the end of the manuscript.

Best regards,
The authors

Round 2

Reviewer 2 Report

The revised manuscript has been carefully improved according to the raised suggestions and comments. I have no other questions and I would like to recommend this paper for publication in the current version.

Reviewer 3 Report

Thanks to the authors for addressing the points of the rebuttal. The paper is acceptable for publication.